# Exploring the Usability of α-MSH-SM-Liposome as an Imaging Agent to Study Biodegradable Bone Implants In Vivo

**DOI:** 10.3390/ijms24021103

**Published:** 2023-01-06

**Authors:** Sana Riyaz, Heike Helmholz, Tuula Penate Medina, Oula Peñate Medina, Olga Will, Yu Sun, Björn Wiese, Claus-Christian Glüer, Regine Willumeit-Römer

**Affiliations:** 1Helmholtz Zentrum Hereon, Max Planck Straße-1, 21502 Geesthacht, Germany; 2Section Biomedical Imaging, Department of Radiology and Neuroradiology University Hospital Schleswig-Holstein Campus Kiel, Kiel University, Am botanischer Garten 14, 24118 Kiel, Germany; 3Institute for Experimental Cancer Research, Kiel University, 24105 Kiel, Germany; 4Lonza Netherlands B.V., 6167 RB Geleen, The Netherlands

**Keywords:** α-MSH, liposomes, implant, fluorescence, Mg-10Gd, ICG, in vivo imaging, FMT

## Abstract

Novel biodegradable metal alloys are increasingly used as implant materials. The implantation can be accompanied by an inflammatory response to a foreign object. For studying inflammation in the implantation area, non-invasive imaging methods are needed. In vivo imaging for the implanted area and its surroundings will provide beneficiary information to understand implant-related inflammation and help to monitor it. Therefore, inflammation-sensitive fluorescent liposomes in rats were tested in the presence of an implant to evaluate their usability in studying inflammation. The sphingomyelin-containing liposomes carrying alpha-melanocyte-stimulating hormone (α-MSH)-peptide were tested in a rat bone implant model. The liposome interaction with implant material (Mg-10Gd) was analyzed with Mg-based implant material (Mg-10Gd) in vitro. The liposome uptake process was studied in the bone-marrow-derived macrophages in vitro. Finally, this liposomal tracer was tested in vivo. It was found that α-MSH coupled sphingomyelin-containing liposomes and the Mg-10Gd implant did not have any disturbing influence on each other. The clearance of liposomes was observed in the presence of an inert and biodegradable implant. The degradable Mg-10Gd was used as an alloy example; however, the presented imaging system offers a new possible use of α-MSH-SM-liposomes as tools for investigating implant responses.

## 1. Introduction

Metal implants are commonly used in bone healing treatments. They are preferred over other materials such as polymers and ceramics for their numerous advantages, including high mechanical strength, durability, and biological compatibility. The most common metallic implant materials are stainless steel, pure titanium, and Ti-based alloys [1]. As an alternative material, biodegradable implants are being intensely studied. They are intended to degrade once they fulfill their function, and a second operation for implant removal is not required [2]. Magnesium (Mg)-based alloys are some of the most promising degrading materials, which have already found their way into the clinic [1]. The mechanical properties of magnesium metal are closer to healthy bone tissue properties compared with other metals, such as the widely accepted titanium [2]. Thus, it is an excellent alternative for non-degradable implants [3].

In vivo trials have shown that Mg degradation enhances bone growth due to increased osteoblastic activity [4,5,6]. Mg-based implants has been successfully developed with applications in in vivo studies and also in long-term clinical studies with promising results in bone healing [7,8]. The mechanical properties of magnesium are advantageous, such as corrosion for implant integration. However, depending upon the composition, an Mg implant can corrode faster with a significant disadvantage of hydrogen gas development and debris formation. The released by-products during corrosion can elicit unbalanced inflammation and lead to implant failure [4]. However, upon alloying with the rare earth (RE) element Gadolinium (Gd), Mg implants show superior mechanical properties with corrosion resistance [9,10,11]. It is prominent that increasing the concentration of Gd up to 10 wt.% improves the corrosion properties of Mg [12,13]. The in vitro studies show good tolerability of Mg-10Gd and low production of inflammatory markers [14]. Mg-10Gd implants also suppress the pro-inflammatory M1 macrophages, leading to decreased inflammation [15] and the induction of calcification [16]. With the help of high-resolution techniques such as 2D small-angle X-ray scattering and X-ray diffraction, it has been indicated that Mg and/or Gd deposit locally within the bone matrix at the site of implantation [17]. Some disadvantages need to be followed when suitable Gd-alloyed implants are studied. The loosening of implants due to various biomechanical factors such as micro-motion or biodegradation leads to the release of debris from the implant [18,19]. The remnants were also found to be surrounded by fibrous and bone tissue [12]. Further, Gd retention in various parts of the human body could be a source of concern [20,21]. However, multimodal ex vivo studies have so far shown that by-products of Gd-rich corrosion stay at the implantation site [8]. In vivo monitoring could be a promising method to test biodegradable metal implants.

The first 1–7 days after implantation show an increase in immune cells [9]. In the first 0–2 days post-injury, neutrophil efflux is followed by macrophage aggregation at the injury site. Macrophages are early participants in the inflammation process, as in vivo studies have shown before [10,11,22]. The innate immune system consists of monocyte lineage macrophages, which are essential in initiating inflammatory reactions to provide a phagocytic response to inflammation [23,24,25]. This immune system activation upon metal implantation can be studied. α-MSH is an anti-inflammatory tridecapeptide cytokine derived from opio melanocortin (MC), the hormonal peptides that modulate macrophage reactivity and attenuate inflammation [25]. Earlier, the SM-liposome was successfully modified with α-MSH peptide, and it was used in macrophage targeting in the inflammatory bowel disease (IBD) murine model [26,27]. The covalently coupled α-MSH peptide binds to melanocyte receptors also found in macrophages. In this study, the performance of the α-MSH coupled liposome was analyzed in a rat implant model. Detecting early immune reactions, which are localized and cell-specific, is pivotal for assessing the fate of bone healing and perhaps predicting implant integration. The crucial factor while developing these types of new bone implant materials is that we need to study the early inflammation pattern in the micro-environment, which includes tissues around implants [28].

The successful initial stages of acute inflammation are essential for bone healing [29]. Optical imaging has been used in preclinical in vivo studies to study implant-related inflammation using bioluminescence in genetic pre-clinical models [30,31]; other studies have assessed either non-invasive, indirect parameters such as animal weight loss or visual inflammation, or direct invasive procedure such as biopsy, for histological examination [32]. Some in vivo markers monitor disease-related inflammatory activity such as collagen-induced arthritis or biomaterial-associated infection in murine models [33,34]. These commercial fluorescent markers, which are available for analyzing early immune reactions, still unfortunately target general or sterile inflammation instead of much-required localized inflammation [35,36]. Non-invasive optical in vivo imaging offers the potential to detect macrophages at the local inflammation site. Fluorescence molecular tomography (FMT) is regularly used in preclinical oncological studies to observe the disease and the efficacy of therapy treatments. It detects near-infrared (NIR) fluorescence and helps in imaging several centimeters inside tissues. This setup is enough for small animal imaging [36]. A cell-specific inflammation marker for in vivo imaging for detecting implantation-related immune reactions would benefit the management of the fate of bone healing [23]. Using liposomes as a contrast agent with a payload of fluorescent dye and armed with a cell-specific ligand to target a cell type can be useful for monitoring inflammation. The macrophage-specific targeting of liposomes may open a way for an image-supervised drug delivery system at the desired site [37].

Over the past 30 years, liposomes have received substantial attention as a pharmaceutical carrier for transporting substances into the body. They have been investigated as potential drug delivery systems. They are biocompatible, in pharmaceutical means, and can stabilize incorporated therapeutic agents. They can also overcome the obstacles to cellular and tissue uptake [38,39]. Their target recognition abilities are limited. However, experimental studies have shown the use of targeting ligands to increase the delivery specificity for diseased tissues and cells [40,41,42]. Our group has developed a liposomal imaging system that consists of sphingomyelin (SM)-liposomes (Figure 1). The SM-liposomes are loaded with indocyanine green (ICG), which is a fluorescent marker used in clinical setups and in optical and photoacoustic imaging [26,43,44]. The SM-liposomes have been shown to target early implant inflammation after surgery, such as the Mg-10Gd implants in mice [45]. In the SM formulation, the liposomes have a cationic surface charge that increases liposome adhesion to the cell membrane. The SM-liposomal implant-targeting correlates well with the commercially available in vivo inflammation marker, which measures MMP activity [33,44]. Here, the authors investigated the possibility of using the α-MSH-SM-liposomes in blood circulation to detect and monitor local inflammation using optical imaging.

The α-MSH-SM-liposomes loaded with an optical ICG tracer were analyzed in vitro using fluorometric assays to study the effect of possible implant-released debris as a long-term implant effect and fluorescent quenching of liposomes by these products or debris. In addition, liposomal intake by bone-marrow-derived macrophages (BMDM) was investigated. Finally, the liposomes were applied to the rats carrying the screw as an implant in tibia. The rats were imaged optically in the early implantation phase to analyze the targeting efficacy, and the performance of α-MSH-SM-liposomes was observed.

## 2. Results

### 2.1. In Vitro Assay for α-MSH-SM-Liposome and Mg and/or Mg-10Gd Interaction

Previous studies have shown that ICG can be encapsulated into the SM-containing liposomes [37,46] and demonstrated that the liposomes target the subcutaneously placed Mg- and Mg-10Gd implants in mice [47]. The present study focused on the interaction between α-MSH-SM-liposome and degradation products of biometal implants in vivo. To better understand liposome behavior in the implant microenvironment, it was studied whether liposomes affect metal degradation. It was also analyzed whether released ions influence the detection sensitivity of the fluorescence marker.

#### 2.1.1. Fluorometric Analysis

In order to determine whether released ions as degradation by-products affect the ICG fluorescence, a test was carried out. The liposomes were treated with variable Mg (Figure 2A) and Mg-10Gd degradation products normalized to the Mg ion concentration (Figure 2B), and the fluorescence from the liposomes was measured. Liposomal fluorescence did not change when the liposomes were exposed to the variable concentrations of free Mg ions (Figure 2A) or Mg and Gd ions (Figure 2B). However, instead of quenching, an increase in intensity was observed in the presence of Mg-10Gd extract at 810 nm (Figure 2B). Based on our measurements, the ions do not have any effect on liposomal ICG fluorescence, and they can be used as an inflammation imaging marker in an environment where ion concentrations increase over time.

#### 2.1.2. Liposome Effect on Degradation of Mg

The liposome immersion study was performed using Mg and Mg-10Gd discs to demonstrate the liposome effect on implant degradation behavior in phosphate buffered saline (PBS) and Dulbecco’s modified eagle medium (DMEM). The degradation was performed, and as a control, Mg discs were studied with SM-liposomes for 24 h and 72 h. Following the in vivo study setup, the Mg-10 Gd discs were studied in the presence of α-MSH-SM-liposome for 24 h and 72 h.

The Mg or Gd ion concentration released in the immersion medium indicates no change in the degradation pattern in the presence of the liposomes for the 24 h and 72 h time ranges (Figure 3). Based on our degradation study, it can be concluded that liposomes do not disturb the implantation process and are suitable as marker molecules.

#### 2.1.3. Bone-Marrow-Derived Macrophages (BMDM)—α-MSH-SM-Liposome Uptake Assay

In previous studies, α-MSH-SM-liposomes were used to target intestine inflammation [27] in order to test their usability on macrophage targeting in the bone implant area. Firstly, the liposomal intake of bone-marrow-derived macrophages was evaluated in vitro. BMDM cells were isolated from the femur and tibia of 6–8 week-old rats; collected cells were plated on a 24-well plate with macrophage medium and grown for 6 days. The purity of BMDM was evaluated using flow cytometry. On day 7, liposomes were added to the cells and incubated for 30 min. After incubation, the cells were washed and fixed with 4% paraformaldehyde. The fluorescent microscopy data show that the negative control without liposome treatment shows only nuclear staining with DAPI (Figure 4A); however, the liposomal ICG uptake of the rat BMDMs can be observed when treated with SM-liposome as well as α-MSH-SM-liposome (Figure 4B and Figure 4C, respectively). Interestingly, the fluorescence microscopy images show that α-MSH coupling changed the liposomal fate in rat macrophage cells. There is partial co-localization of ICG with DNA label (Figure 4C).

### 2.2. In Vivo Evaluation of α-MSH-SM-Liposomes

The tibia implantation in the rats (Figure 5A) was performed to evaluate the liposomes’ fate in vivo. The Mg-10Gd implants were compared to a non-operated and Ti implant group. Ti was chosen as a control along with the non-operated group for comparison because it carries a gold standard as a well-tolerated implant material. After the surgery, the treated legs were visually examined to observe any signs of inflammation such as skin redness (Figure 5B). μCT image scout view was taken for quality control of surgery and confirming the correct implantation of screws (Figure 5C).

The muscle was used as a control for non-specific binding in the non-operated area in rats. The representative image with Mg-10Gd implant (Figure 6) presents the imaging study setup and explains how the ROI is chosen in the in vivo study. The fluorescence data indicated that the maximum signal is obtained at 1 h for all three groups (Figure 7C). There was a non-detectable fluorescence signal without liposome injection 24 h post-surgery. The highest signal was detected 1 h after liposome injection, followed by a signal reduction after 24 h and even more at 72 h. The decline in signals at the 1 h time point shows the non-retention of liposomes at the site of implantation and the surrounding soft tissue. The 1 h background fluorescence analyzed from muscle is the same in all cases and shows that the injection was successfully repeatable in all groups. The fluorescence signal of the non-operated bone is significantly higher at 24 h compared with other groups (Ti, Mg-10Gd). This could be because the operated cases have higher immune cell activity and blood flow. We found a statistical significance using a two-way ANOVA test (Tukey’s multiple comparison test, *p* < 0.05), indicating faster liposome clearance in the operated animals (Figure 7C) without any accumulation at the sites.

### 2.3. Ex Vivo Biodistribution Analysis

At 72 h post-injection, the animals were sacrificed, organs were collected, and signal accumulation at the study end point was measured from organs based on ICG fluorescence in different organs (Figure 8A). Observing macroscopic imaging of excised organs is a valid method for studying the uptake or retention of liposomes in the organs; however, it means that further data cannot be collected. There was a very low ICG fluorescence signal from all the organs obtained from animals of the Mg-10Gd group (Figure 8B). Contrastingly, fluorescence signals could be detected in all organs from the non-operated group. Interestingly, in the Ti-operated group, a high accumulation of ICG fluorescence can be seen in the liver. The results indicate the Mg-10Gd did not retain the signals as compared with the non-operated and Ti-operated group in the organs. A dramatically high signal in the liver of the Ti-operated group is observed. This striking result needs further investigation. One hypothesis could be that the liposomal ICG has been transported to the immune and secretory organs. The speed of clearance represents the inflammatory and immune cell activity. The literature suggests that the liposomes, when circulating in proteinaceous serum, obtain a corona around them; this accelerates opsonization, phagocytosis, and clearance of the liposomes [42,48].

## 3. Discussion

For studying inflammation in the implantation area, non-invasive imaging methods would offer a way to follow the process from the beginning to the healing state. This study does not only measure the inflammation status of the implant area based on the liposomal fluorescent tracer but also reports the lipid nanoparticle delivery potential on inflamed tissue in the near vicinity of the implant. This opens totally new ways to treat the tissue surrounding the implant, while lipid-based delivery has recently been used for the delivery of multiple substances such as anti-inflammatory substances, cancer drugs, and DNA and mRNA. In this study, we used SM-liposomes, which were earlier successfully used in in vivo inflammation imaging. The SM-liposomes were boosted with a targeting moiety, α-MSH, to target the macrophages with these liposomes at the implant site. The purpose of this study was to evaluate the performance of α-MSH-SM-liposomes. The liposome evaluation was started in vitro to analyze if there was any specific interaction between the fluorescent liposomes and implants or implant corrosion products. It is valuable to study the relationship between representative corrosion products from the implants and their potential effect on the quenching of fluorescence of liposomes. The release of debris is a long-term effect caused by implant–host interaction. We studied the direct impact of ions and debris released by the implant on the sensitivity of the liposomal probe without observing any interference (Figure 2). This result highlights that a liposomal fluorescent inflammation probe can be used to evaluate the severity of inflammation and the healing process. Liposomes did not impact the corrosion characteristics of the implants (Figure 3). Thus, they offer a neutral marker method for long-term in vivo studies with biodegradable implants.

The immune reactions with a balance of beneficial responses in bone healing and correct macrophage reactions are essential for bone recovery [49]. It makes sense to target the macrophages to estimate the inflammation in the presence of implants. This targeting could help govern implant material selection. Hence, an appropriate in vivo imaging modality with a suitable imaging agent would help to evaluate different implant materials’ performance and allow the long-term perception of the implant’s biological effects on the body [46]. We wanted to assess the success of targeting the liposomes to the macrophages in the implant site. Earlier, it was shown that monocytes; macrophage subtypes M1, M2; and colon cancer cell line Caco-2 effectively take up SM-liposomes in vitro [50]. However, in bone fraction studies, we wanted to focus on BMDM cells extracted from bone marrow in the rats’ femurs and inducing these BMDMs to macrophage to check the uptake of liposomes. We could see the successful uptake of α-MSH-SM-liposomes on macrophages but we could not define the specificity of liposome targeting successfully to macrophages in this study (Figure 4A–C). Further studies are needed to evaluate the targeting specificity.

The metallic biodegradable implant disintegrates at the site, and its products release into the biological environment over time. The host must tolerate these breakdown products without eliciting an undesirable immunogenic response. The effect of the breakdown product in the body needs to be monitored [8,9,18,22]. In vivo studies would offer an excellent method to study these long-term effects. In in vivo studies (Figure 7A–C), two implant materials were used: a non-biodegradable Ti and a biodegradable Mg-10Gd. When bone tissue was compared to muscle tissue at the early phase of implantation, the ICG fluorescence from the liposomes could be traced in the inflammation site 1 h after liposomal administration. The imaging results indicate the suitability of the liposomes in the bone-implant system, while they were cleared without retention after 72 h. Only the non-operated group gave significantly high signals in the bone at 24 h compared with the other operated groups.

The ex vivo data where ICG fluorescence was analyzed from organs (Figure 8A,B) indicate that the Mg-10Gd implanted group did not retain the signal compared with the non-operated and Ti-operated groups. A remarkably high signal in the liver in the Ti-operated group was observed, indicating that the main secretion route for ICG is through the liver and hepatobiliary pathways, which is typical for liposomes, and no specific targeting probably occurred. The α-MSH-SM-liposomes were cleared from the body faster in the Mg-10Gd implant group. It also showed a smaller signal in the injury and implant area. It is possible that the Mg-10Gd implant has an immunologically smoothing influence, and there are fewer macrophages at the injury site. There is literature that hints at this [15], but more studies are needed on Mg-10Gd implants to elucidate this more.

We tried to focus on the microenvironment around implants, which includes the bone marrow, soft tissue, etc., by performing in vitro experiments to understand the relative effect; however, alike to in vitro, there is a non-significant effect on liposome clearance in vivo. This clearance is not dependent on the implant material used in the study. Irrespective of implant materials, we observed a non-significant effect on liposome activity and its clearance from the animal body.

The system is complex but realistic due to the involvement of multiple factors (imaging, biology of implant–blood interaction, material/foreign body reactions), which are involved in implant-related inflammation, and the challenges that are currently available to us. We were not able to completely define the accuracy of the macrophage targeting using α-MSH-SM-liposomes in our study. Still, we have prepared the ground to investigate further the targeting of immune reactions in the microenvironment of the implant–bone area using optical imaging with a novel liposome formulation.

## 4. Materials and Methods

### 4.1. Preparation of the Implants

Screws were manufactured from extruded Mg alloys with a weight percentage (wt.%) of Gd of 10 wt.%, indicated with Mg-10Gd. The Mg-10Gd screws were cleaned using the standard cleaning procedure. This involves treating the screws serially with n-Hexane (20 min), acetone (20 min), 100% ethanol (3 min), and 70% ethanol (30 min), followed by drying the screws for 60 min. After cleaning, the screws were sterilized using gamma radiation. The corrosion rate in vitro is 0.20 mm/year. The screws were 4 mm long and 2 mm in diameter, with M2 thread. The titanium screws were purchased from Promimic AB (Mölndal, Sweden).

### 4.2. Preparation of α-MSH-SM-Liposome

Lipids were purchased from Avanti polar lipids (Alabaster, AL, USA) and the coupling reagents from Sigma (Darmstadt, Germany). In SM-liposomes, SM:DOTAP:DSPC:Cholesterol: Maleimide-PE (30:20:20:30:0.5 mol%, respectively), a total of 20 μM lipids dissolved in chloroform were pipetted into glass vials, dried under streaming N_2_ gas, and subsequently lyophilized overnight. [Nle4,D-Phe7]-α-melanocyte-stimulating hormone (NDP–α-MSH) (Avanti polar lipids, Alabaster, AL, USA) was coupled to the liposomes using maleimide-PE. This peptide was chosen as it was previously approved to be the most stable version of α-MSH-peptides [38]. The lipid film was hydrated at 60 °C for 30 min with PBS containing 1 mg/mL ICG (Sigma, Darmstadt, Germany) and 0.3 mg/mL carboxyl-coated 5 nm Fe_3_O_4_ with Cathechol-PEG (400)-COOH nanoparticles (A.C. Diagnostics Inc. Fayetteville, AR, USA). The hydration was followed by three freeze–thawing cycles, and unilamellar liposomes were formed using the small-scale LIPEX^®^ Extruder (Burnaby, BC, Canada) with 100 nm pore size membranes (Sigma-Aldrich Chemie, Darmstadt, Germany). The liposomes were purified with PD-10 Sephadex column (G.E. Healthcare, Chicago, IL, USA). The 1.5 mL fractions were collected from the column, and the fractions were analyzed to estimate the encapsulation efficiencies for ICG. After extrusion, the a-MSH-peptide was coupled to the liposomal surface: Melanocyte stimulating hormone trifluoroacetate salt, NDP-αMSH (Sigma, Darmstadt, Germany), was treated with Traut’s reagent 2-Iminothiolane hydrochloride (Sigma, Darmstadt, Germany) to form sulfhydryl (-SH) groups from the primary amines in the peptide. A total 1 mg of the activated NDP-αMSH-peptide was added to the freshly made liposomes (the 1.5 mL fraction) and allowed to react for 30 min at room temperature. Dynamic light scattering size analysis showed that the liposomes with loaded dye had an average size of 194.5 nm [44]. Encapsulation efficiency of ICG to SM-liposomes before αMSH coupling was over 90%.

### 4.3. In Vitro Studies: Material and Liposome Effect on Each Other

#### 4.3.1. Fluorescence Measurement to Analyze the Ion Effect on Liposomal ICG Fluorescence

The liposomal ICG fluorescence was measured in increased Mg and Mg-/Gd concentrations (5 mM, 10 mM, 15 mM, and 20 mM). Mg and Mg-10Gd degradation products or extracts were produced according to EN ISO standards I. 10993-5:2009 and I. 10993-12:2012 (0.2 g material/mL extraction medium) for 72 h under physiological conditions (37 °C, 5% CO_2_, 20% O_2_, 95% relative humidity) [51]. AAS was used to measure Mg and Gd concentration [52]. The extracts were added in liposome dilution (1:25; *v*:*v*, same ratio as the liposome injection to the rat blood) in α-minimal essential medium (MEM with nucleoside, Sigma-Aldrich Chemie, M0450), and the fluorescence was measured using a Tecan Safire Infinite^®^ 200 PRO plate reader 2.0.10.0 (Tecan Group Ltd., Männedorf, Switzerland).

#### 4.3.2. Liposome Effect on Degradation of Mg

An in vitro immersion test calculating the weight loss after 72 h immersion under cell culture conditions (37 °C, 5% CO_2_, 20% O_2_, humidified atmosphere) was performed as described in Nidadavolu [53] in order to determine the impact of liposomes on the degradation of the Mg-10Gd material. Material discs 1.5 mm in height and 1 cm in diameter were cleaned utilizing 20 min sonication in n-hexane (Merck, Darmstadt, Germany), 20 min sonication in acetone (Merck, Darmstadt, Germany), and 3 min sonication in 100% ethanol. Lastly, specimens were sterilized and dried in 70% ethanol under sterile conditions [54,55].

The discs were placed in Dulbecco’s modified eagle’s medium (DMEM, Life Technologies, Darmstadt, Germany) + 10% FBS (fetal bovine serum, PAA Laboratories, Linz, Austria): αMSH-SM-liposome (25:1 *v*:*v*), and in DMEM and DMEM 10% FCS: PBS (phosphate-buffered saline) (25:1 *v*:*v*). The Mg concentration released into the immersion medium was measured utilizing atomic absorption spectrometry (Agilent 240 AA, Agilent, CA, USA), as described [54]. The immersion assay was also performed with pure Mg using the same methodology.

### 4.4. Macrophage and α-MSH-SM-Liposomes Uptake Assay

#### 4.4.1. Preparation of Bone-Marrow-Derived Macrophages In Vitro

Rat bone-marrow-derived macrophages (BMDMs) were isolated from the femur and tibia of 6–8 week old Sprague Dawley rats. Marrow was flushed from the bones using 10 mL cold PBS in 21-gauge needle; cells were centrifuged for 5 min at 300× *g* and re-suspended with macrophage medium (Iscove’s Modified Dulbecco’s Medium (IMDM)), 10% PBS + 1% P/S (Penicillin–Streptomycin 10,000 U/mL), and 10 ng mL^−1^ macrophage colony stimulating factor (M-CSF, Peprotech, Cranbury, NJ, USA). The mixture was centrifuged for 4 min at 300× *g* and collected cells were then plated on a 24-well plate with macrophage medium on sterilized glass slides. The medium was replaced every 2 days during the next 6 days until complete macrophage induction and formation.

#### 4.4.2. BMDM Quality Check with Flow Cytometry

The purity of BMDM was observed using flow cytometry with MACS Quant X (Miltenyi). Data analysis was performed with FCS express research edition 7 (De Novo Software, Pasadena, CA, USA). Staining was performed using phycoerythrin (P.E.) conjugated CD 68 antibody (MCA341PE, Bio-Rad Laboratories GmbH Feldkirchen, Germany), APC-Vio770 conjugated CD 11b (Bio-Rad Laboratories GmbH, MCA275A647, Feldkirchen, Germany), and Viability 405/520 Fixable Dye (130-109-814, Miltenyi, Bergisch Gladbach, Germany). The cells were stained with Viability dye 1 µL per 10^6^ cells for 15 min (as per manufacturer instructions) at room temperature (RT) in the dark. Cells were washed at 300× *g* for 5 min. The supernatant was discarded, and the pellet was washed with FACS buffer. Extracellular staining was achieved by adding 1 µL CD11b antibody to 100 µL (1:100) cell suspension and incubating for 30 min at RT in the dark. Cells were washed at 300× *g* for 5 min. The supernatant was discarded, and the pellet was washed with FACS buffer. Cells were fixed and permeabilized using Leucoperm fix/perm (Bio-Rad Laboratories GmbH, Feldkirchen, Germany) using manufacturer instructions. Intracellular staining for CD 68 was achieved using CD 68 (1:50) antibody by adding 2 µL to 100 µL cell suspension and incubating for 30 min at RT in dark. Cells were washed as described above. Isotype staining of BMDM with IgG1-PE (GM4993, Life Technologies, Darmstadt, Germany) and IgG2-AF 647 (51-4724-81, Life Technologies, Darmstadt, Germany) was performed. After preparation, samples were analyzed for their purity before being processed further for the assay (Appendix A).

#### 4.4.3. Liposome Assay with BMDM

On day 7, liposomes were added (200 µL) to the cells and incubated for 30 min in the incubator (37 °C, 5% CO_2_). After incubation, the supernatant was removed, followed by washing with PBS (Gibco, pH 7.2, Carlsbad, CA, USA). The cells were fixed with 4% paraformaldehyde in the dark at room temperature for 30 min followed by washing with PBS. The cell nuclei counterstaining was done with a DAPI solution final concentration of 0.1 µg/mL. The procedure has been previously demonstrated successfully [50].

### 4.5. Animal Experiments

Nine Sprague Dawley rats were obtained from the Charles River Laboratories (Sulzfeld, Germany) and maintained in a temperature- and humidity-controlled environment, with a 12 h light/dark cycle and access to food and water ad libitum. Adult (aged 8 to 12 weeks) animals of both sexes were used for the experiments.

All procedures were performed according to the guidelines of institutional authorities and approved by the Ethics Committee for Animal Experiments at Christian-Albrechts University of Kiel, Germany (animal testing license no V 241-26850/2017(74-6/17)).

The animals were placed in three groups: non-operated group (*n* = 3), titanium operated group (*n* = 3), and Mg-10Gd operated group (*n* = 3). The Ti and Mg-10Gd implants were inserted into animals with surgery. A dosage of 75 mg/kg Ketamine and 0.5 mg/kg bw Medetomidine (injection volume about 250 µL) were injected intraperitoneally. The knee region of the animal was shaved and treated with a skin disinfectant (Octenisept). The operation was performed by making a longitudinal incision medially along the lower leg 8 mm below the knee joint (approximately 2 mm distal to the knee joint and about 1 mm median to the crest of the tibia) with a scalpel. The underlying muscles were separated along the tibia. The bone was exposed adjacent to the attachment of the patellar tendon. The cortical bone of the tibia was drilled through with a 1.4 mm electric drill (dental micromotor Marathon s04 (N7S)). The screw was then carefully tightened with a hand drill in the cortex. Approximately 2 mm of the screw protruded from the bone, leaving the opposite side of the cortical bone of the tibia undamaged. The wound was sutured with an absorbable suture (5-0, Vicryl USP). Spray plaster was applied to the skin sutured with single button sutures. After the surgery, the animal was placed on the warm plate and partially antagonized (Atipamezole, 1 mg/kg bw, 50 µL subcutaneously). After 2 h of surgery, the animals were again administered Metamizole (100 mg/kg bw, sc, injection volume was approximately 50 µL). The animals were treated with tramadol 72 h after surgery via drinking water (at the dosage of 0.5 mg/mL drinking water [55]).

The α-MSH-SM-liposomes loaded with ICG were injected via intravenous application (10 mM concentration in a 400 µL volume) 24 h post-surgery. The animals were imaged at four different times after surgery: before the liposome injection and 1 h, 24 h, and 72 h after liposome injection.

### 4.6. Imaging of α-MSH-SM-Liposomes

The imaging was performed under anesthesia with 75 mg/kg Ketamine and 0.5 mg/kg Medetomidine by intraperitoneal injection, allowing the animals to be immobilized for 2 h. The animals were injected with 400 µL of α-MSH-SM-liposomes. The anesthetized animals were placed in imaging cartridges of 35 mm height and inserted into the imaging dock. Fluorescent images were acquired using the fluorescent molecular tomography technique from FMT 2500 (PerkinElmer, Waltham, MA, USA) on the channel for 790 nm. The acquisition time was 10 min for the region of interest (ROI), which was defined by observing the area of sutures in the animal leg. The animals were imaged before the marker application (0 h), and after the liposomes were injected at 1 h, 24 h, and 72 h. Fluorescence molecular tomography was used to analyze the liposomal targeting on the site of the implant. The animals were imaged 24 h after surgery before liposomal addition and 60 min after liposomal injection. The fluorescence imaging was repeated after 24 h and 72 h. The ICG fluorescence was analyzed from FMT images. More precisely, a region of interest (ROI) was defined around the implantation site. The images were fitted with an ellipsoidal volume of interest (VOI); after that, the maximal signal within the tibia region and muscle region was calculated by software. The images were analyzed using TrueQuant software version 3.1 (PerkinElmer Inc., Waltham, MA, USA). The animals were sacrificed 72 h post-application of α-MSH-SM-liposomes, and the organs (lungs, liver, kidney, and heart) were collected and imaged immediately for the fluorescent signals from α-MSH-SM-liposomes (Ex/Em: 740 nm/790 nm) using NightOwl II LB 983 by Berthold Technologies (Baden-Württemberg, Germany).

### 4.7. Statistics

Statistical analyses were conducted using the GraphPad PRISM 9 (San Diego, CA, USA) software package for Windows. Due to the non-normal data distribution, the nonparametric one-way ANOVA test was used to evaluate significant differences between samples. Statistically significant differences were considered at *p* < 0.05 as well as two-way ANOVA highly significant changes with *p* < 0.0001.

## 5. Conclusions

Applying α-MSH-SM-liposome formulation in a bone-implant model system is the first insight into immune-related inflammation imaging. The study opens up the option of using liposomes for in vivo imaging of degradable Mg-based metallic implants. Using these liposomes for localized inflammation would be a promising tool for monitoring implant development. Further studies must be performed to verify and examine the specificity of α-MSH-SM-liposomes to target macrophages in vivo.

## Figures and Tables

**Figure 1 ijms-24-01103-f001:**
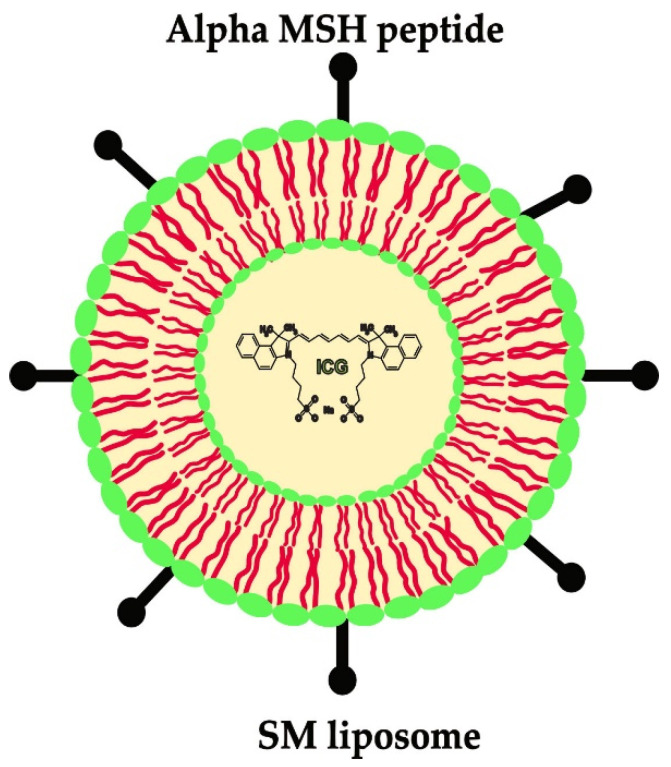
A schematic presentation of the liposomes coupled with α-MSH peptide to make macrophage-specific SM-liposomes. The α-MSH peptide (●▬) is covalently coupled to the lipid membrane. The indocyanine dye is encapsulated in the core as a payload.

**Figure 2 ijms-24-01103-f002:**
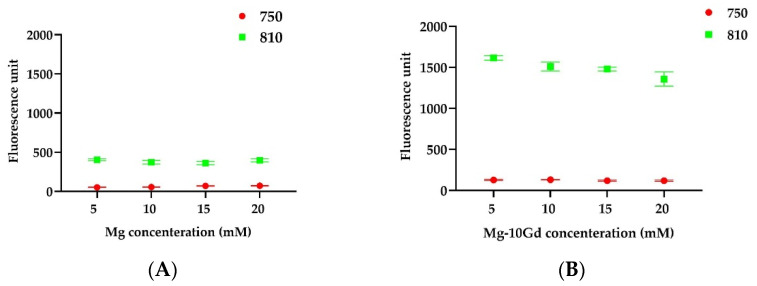
The in vitro assay to study the effect of Mg ions on the liposomal ICG fluorescence was studied. The ICG fluorescence was measured (Ex: 750 nm/Em: 810) from the liposomes loaded with ICG and coupled with α-MSH peptide (● = 750 nm, ■ = 810 nm). (**A**) The liposome dilution (1:25; *v*/*v*) was treated with varying concentrations of Mg (5 mM, 10 mM, 15 mM, and 20 mM). (**B**) The liposome dilution (1:25; *v*/*v*) was treated with varying concentrations of Mg-10Gd (dilution of extract 5 mM, 10 mM, 15 mM, and 20 mM).

**Figure 3 ijms-24-01103-f003:**
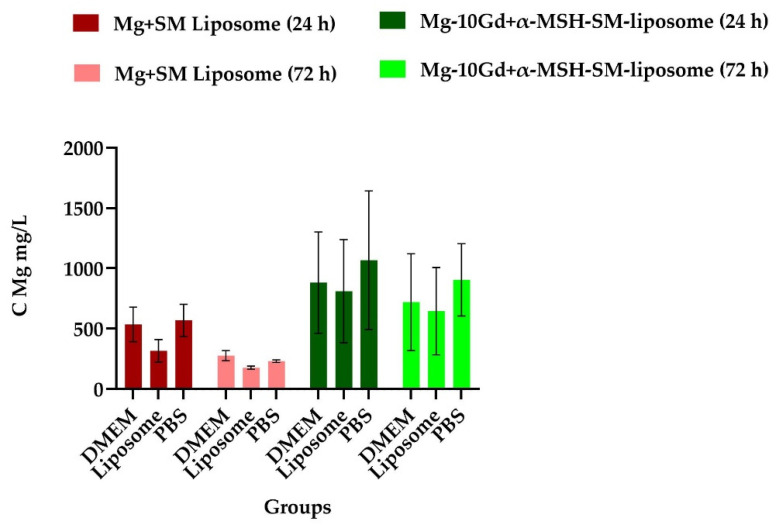
A liposome immersion assay was performed in liposome solution (20 µM, 1:25 dilution in DMEM) to study the degradation rate. DMEM and PBS were used as a control solution. The degradation was performed, and as a control, Mg discs were studied with SM-liposomes. Data are expressed as the mean ± standard error of the six samples per group (*p* < 0.05). The results indicate no change in the degradation pattern in the presence of the liposomes for the 24 h and 72 h time ranges.

**Figure 4 ijms-24-01103-f004:**
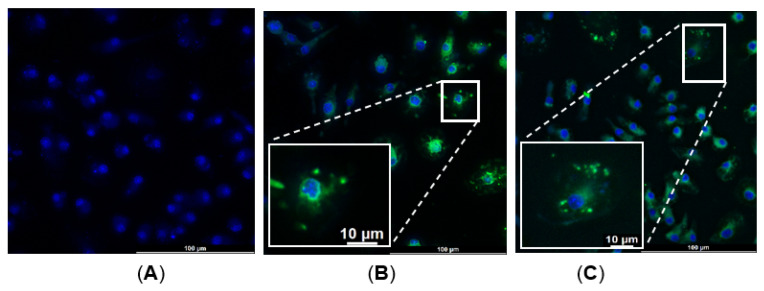
Isolated BMDMs were treated with α-MSH-SM-liposomes or plain SM-liposomes loaded with ICG (green) to study their cellular uptake. The cell nuclei were stained with DAPI (blue). Panel (**A**) represents isolated BMDM cells without liposomes used as a control. The BMDM cells were treated 30 min with SM-liposomes (**B**) or α-MSH-SM-liposomes (**C**); then, they were washed 3 times with PBS to get rid of the non-specific binding. (The images were taken at 63X magnification using Leica 3D thunder microscope, Wetzler, Germany.)

**Figure 5 ijms-24-01103-f005:**
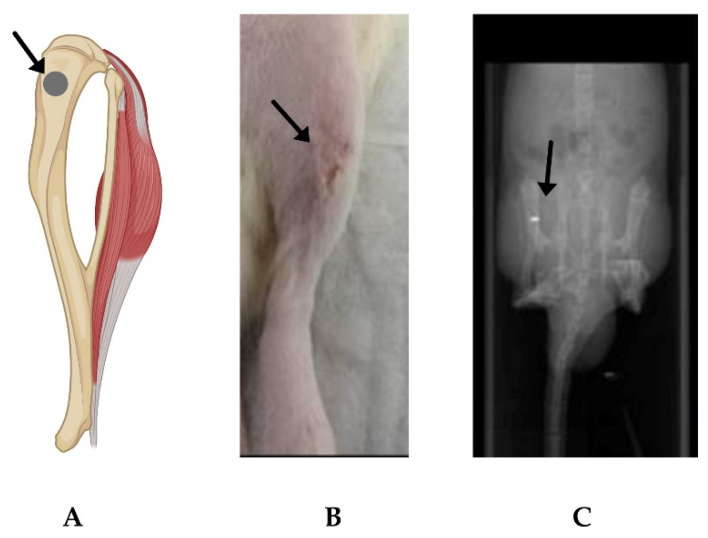
Site of implantation on the animal. (**A**) Diagrammatic representation of the implantation site (arrow) at rat tibia (Created using Biorender.com, accessed on 19 December 2022). (**B**) Representative post-operative image of an animal. (**C**) Representative μCT scout view image for quality control to confirm the implantation site.

**Figure 6 ijms-24-01103-f006:**
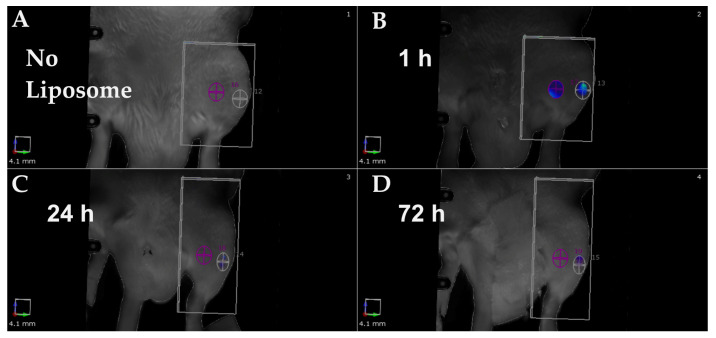
The in vivo imaging of α-MSH-SM-liposome usage in implant inflammation. The imaging was performed before liposome addition and at 1 h, 24 h, and 72 h after it. The panel shows a representative image before fluorescent liposomes were not injected (**A**), post-1 h (**B**), post-24 h (**C**), and post-72 h (**D**) after liposome addition. The ROI was placed at the bone-implantation site with Mg-10Gd (white) and the muscle area (violet/colored) from which fluorescent signals were determined.

**Figure 7 ijms-24-01103-f007:**
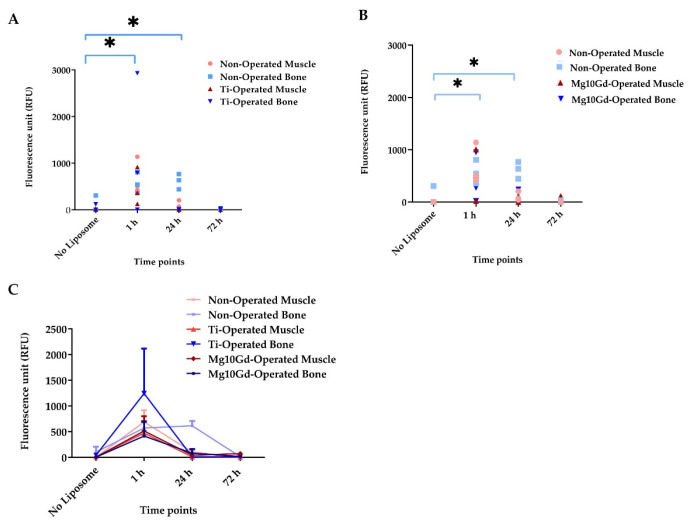
The fluorescence was measured in all three groups using FMT for observing the clearance and signal retention. (**A**) Fluorescence intensities were compared between the non-operated group and the operated group with Ti screw. Data from each animal are presented. (**B**) Fluorescence intensity changes were compared between the non-operated and operated groups with Mg-10Gd screw. Individual animals are presented. (**C**) Fluorescence intensities with the average signal from each group. Data are expressed as the mean ± standard error of the three samples per group; * *p* < 0.05.

**Figure 8 ijms-24-01103-f008:**
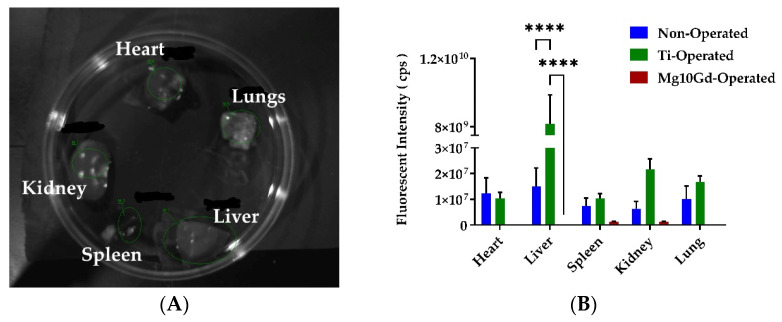
The analysis of ICG fluorescence from ex vivo collected organs to evaluate the biodistribution of α-MSH-SM-liposomes among the three groups. (**A**) Representative image depicting the setup of organs for accumulation or retention analysis of ICG fluorescence. (**B**) Signal retention or accumulation of α-MSH-SM-liposomes in all three treatment groups (non-operated, Mg, and Mg-10Gd) 72 h after liposome injection. ICG fluorescence was measured from heart, liver, spleen, kidney, and lung using NightOwl. Data are expressed as the mean ± standard error of the three animals per group; **** *p* < 0.0001.

## Data Availability

The data presented in this study are available on request from the corresponding author (sana.riyaz@hereon.de and heike.helmholz@hereon.de).

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
