# Peer review of "Exploring the Usability of α-MSH-SM-Liposome as an Imaging Agent to Study Biodegradable Bone Implants In Vivo"

_ijms, 2023, doi:10.3390/ijms24021103_

Round 1

Reviewer 1 Report

Point 1, The loading and encapsulation efficiencies need to be clearly identified. Size zeta potentials are equally required. If reported, basic introduce is necessary.

Point 2, For the targeting experiment, control design is poor. How about the targeting efficiency among cell types or tissues? How about the off-targeting efficiency?

Point 3, How about the toxicity? It was reported liposomes can lead to inflammation response, and how about the effect of liposomes on the macrophages? If the diagnostic agent liposomes can induce inflammation in loci, how can the side effects be avoided? Otherwise, the meaningfulness would be severely discounted.

Point 4, For animal experiment, please clarify the exact ICG mass doses, and the sensitive limits for the purpose of diagnosis. What is the liposome dose? Is the dose safe?

Point 5, Writing issues. Line 118, “have” to “has”. Line 496, before or after? Check other similar issues throughout the text.

Point 6, Fig labelling, scale bars are not clear enough. Check fig 8, fig 1 and fig 4. Fig 5 is blank. Different fonts in figs.  

 Point 7, Introduction section would be better to precisely state the problems of the current strategies and to propose their novel strategy. However, I cannot catch the emphasis of this introduction. It is too tedious.

Author Response

Dear Reviewer 1,

Thank you for your valuable comments on our paper- ‘Exploring the usability of α-MSH-SM-liposome as an imaging agent to study biodegradable bone implants in vivo.’ The suggestions have been helpful, and we also appreciate your insightful comments on revising the introduction and other aspects of the paper.

We have revised our manuscript as suggested and present the detailed answers to all comments below.

Sincerely,

Sana Riyaz

Response to Reviewer 1 Comments

Point 1: The loading and encapsulation efficiencies need to be clearly identified. Size zeta potentials are equally required. If reported, basic introduce is necessary.

Response 1: The loading and encapsulation efficiencies with size zeta potentials had been measured earlier (Ref: https://www.eurekaselect.com/article/108477 ), with an average size of 194.5 nm.

Lines 419-421 have been added in the material methods of the manuscript (4.2: Preparation of α-MSH-SM-liposome): “Dynamic light scattering size analysis showed the liposomes with loaded dye had an average size of 194.5 nm (Ref: https://www.eurekaselect.com/article/108477 ). Encapsulation efficiency of ICG to SM-liposomes before α MSH coupling was over 90%.”

Point 2: For the targeting experiment, control design is poor. How about the targeting efficiency among cell types or tissues? How about the off-targeting efficiency?

Response 2: We have taken two controls, one with no liposome treatment and the second with only SM liposome treatment. We have previously done a study (ref: https://www.ncbi.nlm.nih.gov/pmc/articles/PMC8962329/ ) that demonstrated the uptake of SM-liposomes in inflamed intestinal tissue along with M1 and M2 subtypes of macrophage uptake. Therefore, we only observed the uptake here with bone marrow derived macrophages as we wanted to observe the same tissue interaction similar to our animal studies. From animal studies, a rough estimate of the tissue targeting can be made, but finer tissue targeting and targeting efficiency analyses are beyond this study's scope. However, this type of study in the future should and will be done as mentioned in the discussion section.

Point 3: How about the toxicity? It was reported liposomes can lead to inflammation response, and how about the effect of liposomes on the macrophages? If the diagnostic agent liposomes can induce inflammation in loci, how can the side effects be avoided? Otherwise, the meaningfulness would be severely discounted.

Response 3: Our experiment, “Bone marrow-derived macrophages (BMDM) - α-MSH-SM-liposome uptake assay”, has proven that the use of liposomal formulation is not toxic to the cells. This liposome formulation contains a conjugation peptide of α-MSH that on contrary exhibits anti-inflammatory properties (Ref: https://pubmed.ncbi.nlm.nih.gov/25140322/ ). We have not observed SM-liposome related acute toxicity in our studies.

Point 4: For animal experiment, please clarify the exact ICG mass doses, and the sensitive limits for the purpose of diagnosis. What is the liposome dose? Is the dose safe?

Response 4: The LD50 for ICG is 50-80 mg/kg for animals (Ref:  https://www.drugs.com/pro/indocyanine-green.html ), and the concentration of liposomes is 10mM, and these rats weighed 300g. We have injected 400 uL containing 4 μmol of lipids with 1mg/ml ICG dose. The average of total dosage administrated is roughly 1,3 mg/kg of ICG, which is forty times smaller than LD50. For humans, the dose per kilo would be several folds smaller, as normally is in the case of imaging tracers. And also, If we think of the lipid dose as well,  it is below 10 mg/kg, and it has been shown that there is hardly any measurable effect below 100mg/kg in murine. K.B. Knudsen et al / Nanomedicine: Nanotechnology, Biology, and Medicine 11 (2015) 467–477 (Lines 495-497)

Point 5: Writing issues. Line 118, “have” to “has”. Line 496, before or after? Check other similar issues throughout the text.

Response 5: The “have” is changed to “has” in Line 48, 52, 159 of manuscript. In Line 513-514, the line is changed to “and after 1 h, 24 h and 72 h of liposome injection”.

Point 6: Fig labelling, scale bars are not clear enough. Check fig 8, fig 1 and fig 4. Fig 5 is blank. Different fonts in figs. 

Response 6: Figure 1 is corrected with clear labeling in the image. Figure 4: labeling has been corrected with scale bars. In Figure 5 the blank box was created while uploading, but now it has been removed. The font has also been corrected in Fig 7 and 8, it is made similar size palantino Lino type with size 10.

Point 7: Introduction section would be better to precisely state the problems of the current strategies and to propose their novel strategy. However, I cannot catch the emphasis of this introduction. It is too tedious

Response 7: This study laid a foundation for future bone inflammation imaging. The system is complex but realistic due to the involvement of multiple factors (imaging, biology of implant-blood interaction, material/foreign body reactions) involved in implant-related inflammation, and the challenges currently available to us. Although it is long, all the mentioned points are essential to give the reader a complete overview for an explanation of the importance of the novel strategy. This study does not only measure the inflammation status of the implant area based on the liposomal fluorescent tracer. It also reports the lipid nanoparticle delivery potential on inflamed tissue in the near vicinity of the implant. It opens totally new ways to treat the tissue surrounding the implant, while lipid-based delivery has recently been used for the delivery of multiple substances like anti-inflammatory substances, cancer drugs, and DNA and mRNA.

Lines added 380-383: “The system is complex but it is realistic due to the involvement of multiple factors (imaging, biology of implant-blood interaction, material/foreign body reactions) which are involved in implant-related inflammation, and the challenges that are currently available to us.”

In response to the answer for reviewer 2, lines 118-124 are also added.

Reviewer 2 Report

In this manuscript, Riyaz and co-workers reported the development of an inflammation-sensitive liposomal platform labeled with an alpha-melanocyte-stimulating hormone (α-MSH)-peptide. The interactions between the resulting liposomes and Mg-based implants (Mg-10Gd) were carefully studied in vitro and in vivo, and no interference was found. As a follow-up study from their group, this report provides a promising liposomal system that can be employed in monitoring implant development. The manuscript is well-written, and the experiments are carefully designed and carried out. The reviewer may suggest some minor changes to the current version of the manuscript.

1.     While the introduction section provides a sufficient description of some of the aspects discussed in the manuscript, the introduction of liposome as a delivery vehicle is not mentioned enough. The reviewer thinks adding that information in the introduction could be helpful.

2.     In lines 154-155, a “vivo” seems to be missing in the sentence.

3.     In Figure 4, the authors pointed out some partial localization of ICG fluorescence and DNA label. It is ideal to do some statistical analysis (i.e., R-value) to prove this. 

Author Response

Dear Reviewer 2,

Thank you for your valuable comments on our paper- ‘Exploring the usability of α-MSH-SM-liposome as an imaging agent to study biodegradable bone implants in vivo.’ They have been beneficial and insightful.

We have revised our manuscript as suggested and below the comments presented with answers point by point. We thank you for your interest and suggestions.

Sincerely,

Sana Riyaz

Response to Reviewer 2 Comments

Point 1: While the introduction section provides a sufficient description of some of the aspects discussed in the manuscript, the introduction of liposome as a delivery vehicle is not mentioned enough. The reviewer thinks adding that information in the introduction could be helpful.

Response 1: In the introduction section, the liposome as a delivery vehicle is mentioned, and we added: “Over the past 30 years, liposomes have grabbed substantial attention as a pharmaceutical carrier for transporting substances into the body. They had been investigated as potential drug delivery systems due to their biocompatible nature and stabilized in-corporation as a therapeutic agent. They also overcame the obstacles to cellular and tissue uptake [35, 36]. But the target recognition abilities are limited and require more attention. However, experimental studies have shown the use of targeting ligands to increase the specificity of delivery at diseased tissue and cells [37-39]” (Lines 118-124).

  1. Shade, C.W. Liposomes as Advanced Delivery Systems for Nutraceuticals. Integr Med (Encinitas) 2016, 15, 33-36.
  2. Abu Lila, A.S.; Ishida, T. Liposomal Delivery Systems: Design Optimization and Current Applications. Biol Pharm Bull 2017, 40, 1-10, doi:10.1248/bpb.b16-00624.
  3. Ferrari, M. Nanovector therapeutics. Curr Opin Chem Biol 2005, 9, 343-346, doi:10.1016/j.cbpa.2005.06.001.
  4. Puri, A.; Loomis, K.; Smith, B.; Lee, J.H.; Yavlovich, A.; Heldman, E.; Blumenthal, R. Lipid-based nanoparticles as pharmaceutical drug carriers: from concepts to clinic. Crit Rev Ther Drug Carrier Syst 2009, 26, 523-580, doi:10.1615/critrevtherdrugcarriersyst.v26.i6.10.
  5. Sercombe, L.; Veerati, T.; Moheimani, F.; Wu, S.Y.; Sood, A.K.; Hua, S. Advances and Challenges of Liposome Assisted Drug Delivery. Front Pharmacol 2015, 6, 286, doi:10.3389/fphar.2015.00286.

Point 2: In lines 154-155, a “vivo” seems to be missing in the sentence.

Response 1: In lined 162, a “vivo” is added at the end of the sentence.

Point 3: In Figure 4, the authors pointed out some partial localization of ICG fluorescence and DNA label. It is ideal to do some statistical analysis (i.e., R-value) to prove this.

Response 3: Indeed, it is ideal to do the statistical analysis, but the authors present an overall uptake analysis. The study showed that the cellular fate of ICG is a very interesting topic and definitely needs more studies in the future.

Round 2

Reviewer 1 Report

The present form of the manuscript is acceptable, except multiple fonts used in the figs and too long introduction section. 

Author Response

Dear Reviewer 1,

Thank you for your valuable comments on our paper- ‘Exploring the usability of α-MSH-SM-liposome as an imaging agent to study biodegradable bone implants in vivo. ’ The suggestions have been helpful and made the manuscript more readable.

We have revised our manuscript as suggested and present the detailed answers to all comments below.

Sincerely,

Sana Riyaz

Response to Reviewer 1 Comments

Round 2

Point 1: The present form of the manuscript is acceptable, except multiple fonts used in the figs and too long introduction section. 

Response 1: In order to make the introduction less tedious, certain repetitive information has been deleted and the introduction has been restructured to make the information more inter-connected. But keeping in mind the multi-disciplinary nature of this research, the authors would like to request to keep the current information for the readers to understand the various aspects of this research. Here, we are using liposome for imaging inflammation using optical imaging in the presence of a degradable implant. Therefore, it is a multifactorial study, where we are trying to lay the foundation for a question that is indeed complex.

The changes performed in the round 2 revision are as follows:

  • In Line 15 Current address of the author is updated: Lonza Netherlands B.V. 6167 RB Geleen, Netherlands
  • Line 25: will help is deleted and help to be added
  • Line 28-30 (edited): The liposome interaction with implant material (Mg-10Gd) was analyzed with Mg-based implant material (Mg-10Gd) in vitro
  • Line 42: Ti is added
  • Line 40: “due to” is deleted and “for” is added
  • Lines 51-54: partly deleted
  • Line 54: “with application in in vivo studies and also” is added
  • Line 57: deleted “of the implant at the implantation site in bone healing”
  • Line 66: “quick resolution” is deleted and “decreased” is added
  • Line 69-70: added, “Some disadvantages need to be followed when suitable Gd-alloyed implants are studied.
  • However, is added in line 68
  • Lines 72-198: are rephrased with omission of repetitive information
  • Line 205: experiments, including is deleted and using is added
  • Lines 211-212: “and the performance of α-MSH-SM-liposomes was observed” is deleted
  • Figure 1-8, font size is fixed palantino type, size 12, bold regular.
  • Figure 5A is redrawn to make readers more clear about the site of implantation using Biorender.com, which is mentioned in Line 250
  • References have been re-arranged.